# Using Self-Supervised Learning of Birdsong for Downstream Industrial Audio Classification

**Patty Ryan** [1]  **Sean Takafuji** [1]  **Chenhao Yang** [1]  **Nile Wilson** [1]  **Christopher McBride** [2]

## Abstract

In manufacturing settings, workers rely on their sense of hearing and their knowledge of what sounds correct to help them identify machine quality problems based on the sound pitch, rhythm, timbre and other characteristics of the sound of the machine in operation. Using machine learning to classify these sounds has broad applications for automating the manual quality recognition work currently being done, including automating machine operator training, automating quality control detection, and diagnostics across manufacturing and mechanical service industries. We previously established that models taking input pitch information from music domains can dramatically improve classification model performance on industrial machine audio leveraging a pretrained pitch model.

In this work, we explore the use of self-supervised learning on pitch-intensive birdsong rather than a pre-trained model. To reduce our reliance on a pretrained pitch model and reduce the quantity of labeled industrial audio required, we implement self-supervised representation learning using plentiful, license-free, unlabeled, pitch-intensive wild birdsong recordings, with audio data augmentation to perform classification on industrial audio. We show that: 1. We can preprocess the unlabeled birdsong data sample with unsupervised methods to eliminate low signal sample and mask low frequency noise leaving just desirable chirp-rich sample. 2. We can identify effective representations and approaches for learning birdsong pitch content by comparing select self-supervised pretext task training of temporal sequence prediction and sequence generation. 3. We can identify effective augmentation methods for learning pitch through comparison of the impact of a variety of audio data augmentation methods on self-supervised learning. And 4. Downstream fine-tuned models deliver improved performance classifying industrial motor audio. We demonstrate that motorized sound classification models using self-supervised learning with a dataset of pitch intensive birdsong, combined with select data augmentation, achieves better results than using the pre-trained pitch model.

## 1. Introduction

We were introduced to the challenge of classifying industrial audio last year when working with a manufacturer that sought to improve welding quality. The correct distance of the welding device to the weld is a critical element in creating a quality weld. If the weld were to be conducted too close or too far from the weld, the weld would be weak and could fail. The master welder, in a tour of the factory floor, was able to immediately call our attention to the difference in the sounds of good welds and the sounds of bad welds. They had distinctively different pitches due to the reflection of the sound off the surface at different distances. However, the light emitted during welding made photography at this distance impractical. And so we investigated classifying the audio based on representations of the pitch with the immediate application of enabling training by allowing welders to get immediate feedback on the quality of their welds.

While there were recent advances using deep learning in areas of music machine learning classification and music synthesis, there are very few applications of these frequency and pitch machine learning methods on classification of audio in the industrial environment. We leveraged the CREPE pretrained pitch estimation model (Kim et al., 2018) and found it performed reasonably well at classifying weld pitches. We implemented a multi-input ConvNet model combining 1D representations of CREPE pitch estimations from the time domain waveform, and Constant-Q (CQT) 2D transforms of the waveform, yielding a high accuracy classification of welding distance (Ryan et al., 2019).

[1]Microsoft, Redmond, Washington, USA [2]ADM Associates, Reno, Nevada, USA. Correspondence to: Patty Ryan <Patty.Ryan@microsoft.com>.

*Published at the workshop on Self-supervision in Audio and Speech at the 37th International Conference on Machine Learning*, Vienna, Austria. Copyright 2020 by the author(s).

We experimented with other industrial audio datasets with the same modeling approach to understand whether the approach was generalizable. We classified correct and incorrect machining lathe settings and distinguished between the motors of different ferry boats operating on the Puget Sound. However, we had two challenges. Labeled data in industrial audio is scarce and expensive to collect, and relying on the CREPE model prediction proved too slow in industrial production settings. In this work we explore the use of self-supervised learning to reduce our labeled data requirements, and we explore whether we can learn enough of the pitch information from a birdsong dataset to allow us to eliminate our reliance on the CREPE pretrained pitch model.

## 1.1. Contribution

We make several contributions that we outline here. We demonstrate the efficacy of using self-supervision with a pitch-intensive birdsong model to allow downstream classification of pitch-intensive industrial motor audio. The application of learning pitch is extensively applied in the music realm (Kim et al., 2018), (Huang et al., 2018), (Engel et al., 2019). Enabling learning from the pitch present in birdsong, and using those pretrained weights to improve classification on the pitch present in industrial audio is novel. We describe unsupervised methods to preprocess the unlabeled birdsong audio to exclude low quality samples leaving us only with high quality samples. Finally, we demonstrate the efficacy of several audio data augmentation methods at enhancing self-supervised learning of pitch and demonstrate this on the downstream classification task. The source code for the implementation of our paper is available at: https://github.com/SingingData/birdsong-self-supervised-learning

## 2. Data Augmentation for Self-Supervised Learning on a Birdsong Dataset

### 2.1. Dataset

In the foothills of the Carson Range, within the Sierra Mountains, we captured footage and audio from a motion-activated wildlife camera. The camera was trained on bird feeders and the surrounding area, and captured eleven second video and audio samples (44.1 kHz). The birds recorded included Quail, Blue Jays, Black Headed Grosbeaks, Doves, Robins, Red Finches, Stellars Jays, Black-billed Magpies, Yellow Warblers and Varied Thrush, among others. We extracted the audio from the captured video and resampled it to a 22 kHz sample rate.

## 2.2. Preprocessing

Some of our video samples were undesirable and needed to be excluded from our sample. For example, some of the motion activated video samples had inadvertent wind activations with no birdsong. Some samples had very faint birdsong. Still others had background noise including sprinklers, cars, and airplanes. First, we eliminated audio samples with little differential between average magnitude and maximum magnitude of the audio signal. Next, we performed a K-means cluster analysis on the CQT unrolled vectors to quickly identify and eliminate clusters of undesirable noise. These two methods allowed us to quickly eliminate one-third of our sample, leaving 1,252 total clean, high quality audio samples.

## 2.3. Transform

Once these samples were cleaned, we converted pitch and timbre through frequency domain changes over time. We use a Constant-Q Transform (CQT) to a 2D CQT spectrogram for each of our audio waveform inputs. CQT is a time-frequency analysis method with greater frequency resolution at lower frequencies and greater time resolution towards higher frequencies better capturing human audible pitch and timbre. Our use of CQT was inspired by TimbreTRON (Huang et al., 2018).

## 2.4. Augmentations

The data augmentation methods we applied are as follows:

- **Pitch Shifting:** The pitch shift augmentation is applied using the Python library librosa (McFee et al., 2020) with the values {-2, -1, 1, 2} being empirically chosen based on the methods from (Salamon & Bello, 2016). The raw frequency values are shifted in increments of semitones with a positive value increasing the pitch and a negative value decreasing the pitch.

- **Octave Shifting:** The octave shift augmentation uses the same methodology as the pitch shift augmentation with an octave shift of 1 being equivalent to a pitch shift of 12 semitones. We reason that for our pretext task on birdsong data to translate well to the industrial audio setting, very large shifts in pitch would be valuable. We used octave shifts with the values {-2, -1, 1, 2}.

- **Time Stretching:** The time stretching augmentation extends or compresses the waveform by the following rates {2, 5, 0.2, 0.5}. A rate of 2 will lead to the audio sample being twice its original speed, leading to a compressed waveform. Likewise, a rate of 0.5 will lead to the audio sample being half its original speed, creating an extended waveform.

*Table 1.* Classification accuracies on the pretext task with birdsong data. All models trained for 20 epochs.

| ARCHITECTURE | AUGMENTATIONS | TRAINING SAMPLES | TRAIN (ACC) | TRAIN (LOSS) | VAL (ACC) | VAL (LOSS) |
|---|---|---|---|---|---|---|
| TRIPLET ALEXNET | NONE | 763 | 92.27 | 2.5789 | 83.44 | 2.4384 |
| TRIPLET ALEXNET | PITCH + OCTAVE | 11445 | 85.64 | 1.5740 | 81.05 | 1.2945 |
| TRIPLET ALEXNET | TIME STRETCHING | 3052 | 84.19 | 1.5734 | 74.13 | 1.3982 |
| TRIPLET ALEXNET | SPECAUGMENT | 3052 | 87.13 | 3.0731 | 77.16 | 2.9691 |

- **SpecAugment:** Introduced for speech recognition, (Park et al., 2019) applied a frequency mask and a time mask on top of the log mel spectrogram representation of the audio sample. We use the library nlpaug (Ma, 2019) to apply this augmentation on the CQT representation of the audio sample. Using the notation and descriptions from (Park et al., 2019), on each audio sample we apply a frequency mask that covers 30 consecutive frequency channels denoted as $[f, f + 30)$ where $f$ is chosen from a uniform distribution of $[0, \nu - f)$ where $\nu$ is the number of frequency channels in the CQT representation. Additionally, two time masks are applied on 10 and 20 consecutive time steps denoted as $T_0 = [t_0, t_0 + 20)$ and $T_1 = [t_1, t_1 + 10)$ with the additional constraint that $T_0 \cap T_1 = \emptyset$.

## 3. Self-Supervised Learning Methods

### 3.1. Self-Supervised Learning Pretext Task

For the self-supervised pretext task, we chose verifying sequence temporal order, drawing inspiration from the "*Shuffle and Learn*" pretext task by (Misra et al., 2016). We reasoned that the pattern of the birdsong could be learned in order to determine temporal order, and in so doing would enable the pitch of the notes of the birdsong to be learned. We first created tuples of sequences by splitting each sample into four sequence chunks of 2.6 seconds a piece that we denote as (a, b, c, d) following the (Misra et al., 2016) approach. Next, for each sample we labeled a positive example as the sequence (a, b, c) leaving out the last chunk. To create negative examples, we incorrectly ordered the sequence using the left out chunk 'd' resulting in the sequence (b, a, d) and (d, a, b).

### 3.2. Model Architecture

Again, following the "*Shuffle and Learn*" design, we designed a Triplet Siamese network for sequence verification. We reduced the last dense layer of the AlexNet architecture modestly to fit available computational resources. We applied the Lecun normal initializer, leaky ReLu and liberally applied drop-out. We balanced the datasets.

### 3.3. Downstream Task

For our downstream task, we classified Washington State Ferry recordings, distinguishing between the Wenatchee and the Tacoma motors based on 2.6 second samples.

For our downstream architecture, we took just one of the Siamese Triplets to form the basis of our downstream model. We loaded the pre-trained weights on each of the convolutional layers, and added two trainable dense layers and an output layer. We froze the first three layers and allowed the last three layers to be trainable. We trained the downstream task for 20 epochs with each of the data augmentation permutations.

## 4. Results

Self-supervised training on birdsong proved effective for improving our downstream classification model performance. First, two data augmentation techniques in particular, pitch shifting and time stretching, proved the most effective at improving downstream performance. With either of these data augmentation techniques present, our downstream model achieved 100% classification accuracy with 10 epochs of training. By contrast, without pre-training, the downstream model failed to learn. In comparison, the model attained a comparable accuracy of 99.75% using a pre-trained pitch model, CREPE, combined with CQT with spec augment data augmentation. The performance of the model on the pretext task is noted in Table 1. The performance of the model on the pretext task is noted in Table 2 . We note the quantity of augmented training data in the table. For pitch + octave augmentations and our time stretching augmentations, we generated a greater number of training samples which may have resulted in lower training loss on the downstream ferry audio training.

## 5. Related Works

**Self-Supervised Learning**. Self-Supervised methods have shown promising growth in the natural language space involving audio waveforms with recent contributions such as Audio ALBERT (Chi et al., 2020). In the general audio space, there has been a larger focus on learning high-quality audio representations through unsupervised methods such

*Table 2.* Classification accuracies on the downstream task with ferry data. Pre-Trained indicates that the self-supervised model weights were transferred onto the classifier. The augmentations where indicated were applied on the data for the pretext task (the birdsong) but were not applied on the downstream task (the ferry sound). Training Samples: 167 recordings. Validation Samples: 66 recordings.

| ARCHITECTURE | PRE-TRAINED | AUGMENTATIONS | TRAIN (ACC) | TRAIN (LOSS) | VAL (ACC) | VAL (LOSS) |
|---|---|---|---|---|---|---|
| ALEXNET | NO | NONE | 56.29 | 17843.1402 | 53.79 | 943.1954 |
| ALEXNET | YES | NONE | 59.88 | 103428.4201 | 87.12 | 6730.2180 |
| ALEXNET | YES | PITCH + OCTAVE | 74.85 | 19469.7310 | 100 | 0.3400 |
| ALEXNET | YES | TIME STRETCHING | 69.76 | 16478.3154 | 100 | 0.3400 |
| ALEXNET | YES | SPECAUGMENT | 59.28 | 15824.3084 | 92.42 | 256.0956 |

as using autoencoders (Roche et al., 2018) equipped with convolutional layers or additionally with recurrent layers as well (Meyer et al., 2017), (Chung et al., 2016). One self-supervised task (Tagliasacchi et al., 2019) is called TemporalGap which focuses on estimating the length of a time masked temporal slice. Instead of using TemporalGap as our pretext task, we incorporated this task as part of our augmentations through SpecAugment (Park et al., 2019) which allows us an additional augmentation method that has demonstrated crucial value to the quality of the learned representations.

**Audio Representations and Augmentations**. The usage of different transformations on the audio waveform such as the short-time fourier transform (STFT), linear and log mel spectrograms, and continuous wavlet transform (CWT) has been studied on environmental audio classification tasks UrbanSound8K by (Huzaifah, 2017). Additionally from the speech recognition space, there is (Nguyen et al., 2019)

**Application in Vision**. While the focus of our methods is strictly focused on learning from the audio waveform, the method that we drew inspiration from (Misra et al., 2016) is performed on video frames. Other methods for self-supervision when video frames and audio waveforms are available have been explored (Alwassel et al., 2019), (Korbar et al., 2018). Our method applied with "*Shuffle and Learn*" (Misra et al., 2016) offers a new self-supervised learning task to the combined video and audio space.

## 6. Discussion

For pitch-intensive downstream classification tasks, it appears pretraining with license-free birdsong recordings is effective at improving performance, even for modestly sized labelled data sets. For our industrial enterprise implementations of audio machine learning, self-supervised learning is a promising approach. In this case, classification on the ferry motor dataset may be too easy, and we look forward to extending our experimentation to other more challenging industrial audio datasets. We believe audio and video of

the natural world with relevant characteristics may prove a cost-effective data source to build self-supervised learning.

Further experimentation is called for given the differences in our train and validation accuracies, as shown in Table 2. We trained our downstream ferry classification models for 10 epochs each. However, additional training may improve the results.

## 7. Conclusion

In this paper, we share a simple insight into the strong pitch component shared by birdsong, music, and industrial audio. We demonstrate the efficacy of a selection of audio data augmentation techniques at representing the pitch component of birdsong and industrial audio. Additionally, we demonstrate unsupervised data pre-processing methods that allow selection of unlabeled birdsong data to yield pitch-intensive sample suited for self-supervised training. Finally, we demonstrate the effectiveness of using self-supervised learning techniques with a pretext task of sequence temporal order verification at learning pitch information that dramatically improves downstream model industrial audio classification tasks.

In future work, we aim to expand upon our method by leveraging other sources of audio data for the pretext task such as AudioSet (Gemmeke et al., 2017). Additionally, we would like to investigate how our learned representation can be used in conjunction with representations obtained from other pretext tasks such as (Tagliasacchi et al., 2019) to capture different features of the audio waveform.

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
