# OpenReview forum: "Using Self-Supervised Learning of Birdsong for Downstream Industrial Audio Classification"
_ICML.cc/2020/Workshop/SAS — SAS 2020_

### Official Review · AnonReviewer1 · 2020-06-22
**Great applied paper**

**Rating:** 10
**Confidence:** 4

**Review:**

Notes:
  -Sound classification could be used to study machines.  Could also be used for birdsong!

  -Very strange idea: learn on unlabeled bird song recordings, and use the features to learn on small volumes of industrial audio.

  -I greatly enjoyed the introduction, which clearly motivated the task.

  -Recognizing re-orderings is used as the self-supervised task.

Comments:
  -Table 2 should show both training and validation accuracies and losses on the ferry task.  Right now I assume it's showing validation.  For example, the paper claims that the one without augmentation "doesn't learn" but I suspect that it does fit the training set but overfits horribly?  This should be clarified.

  -It seems like good ferry accuracy can be obtained with the augmentations + pre-training.  Also from the paper it's implied that the augmentation is only used while fine-tuning on the ferry data.  Is that right?

Review:
This is a clever use of transfer learning from birdsongs to industrial sound classification.  The experiments could be more thorough and the tables should show more values (such as train vs. test), but on the whole I think this is the kind of applied work that we need to see a lot more of in the self-supervised and transfer learning communities.

---

### Official Review · AnonReviewer3 · 2020-06-23
**Interesting use of data.  Semi-supervised pre-training makes this potentially scalable**

**Rating:** 8
**Confidence:** 4

**Review:**

This paper exhibits a creative data trick. ConvNets for an industrial audio task (distinguishing the motors of different ferries) are pre-trained on a different audio dataset: birdsongs.

What makes this a potentially very high-impact paper is the use of unsupervised methods even in the pre-training.  Pre-training uses the shuffle-and-learn method, predicting correct versus incorrect ordering of subsegments.  In this way, no manual labeling of the birdsongs is required, except only that the start and the end of each birdsong have been segmented.   In this way, it seems like it should be possible to rapidly acquire a reasonably large birdsong corpus.

Methodology is reasonable.  Accuracies on both the birdsong and ferry corpora are reported using development test data, but since there are only five variant systems reported (with widely varying accuracies: 54% up to 100%!), the experimental results are probably valid.  Data augmentation methods harm performance on the birdsong pre-training corpus, but help performance on the ferry fine-tuning corpus --- it would be interesting to explore the reason for this discrepancy, e.g., do the data augmentation methods make the classifier less specific to birdsongs?

The abstract claims that "self-supervised learning ... achieves better results than using the pre-trained pitch model," but this result does not seem to be reported in Table 2 or in the text.

---

### Official Review · AnonReviewer2 · 2020-06-29
**Interesting application of domain transfer via pre-training & fine-tuning**

**Confidence:** 4
**Rating:** 7

**Review:**

This paper describes an approach using AlexNet and triplet loss to pretrain a network to learn a sequential representation of bird songs.  This pretrained model is then fine tuned to distinguish ferries (an industrial application). The only thing these two tasks have in common is that they share a distinguishing features, namely pitch, in (likely) making effective decisions.

Strengths
* This is an interesting domain transfer application.  Often we see tasks that are more obviously related on the semantic or label dimension.  This approach takes the view that since pitch matters to both tasks, a robust representation will help both.  This appears to be correct.
* The sequence based pretraining approach is sensible and effective.

Weaknesses
* The Ferry classification task may be too easy.  While the baseline shows only 55% accuracy, after pre-training and some effective (but standard) data augmentation, performance hits 100%.

Questions:
* The Ferry Loss seems to be very far from the range of the loss during pre-training. While this is to be expected, how do you describe the range observed in table 2, where pitch and octave augmentation can reduce the loss from ~20000 to less than 1.

---

### Decision · Program_Chairs · 2020-07-01

**Decision:**

Accept

**Comment:**

Dear author(s),

Thank you very much for your submission at the ICML2020@SaS workshop (https://icml-sas.gitlab.io/). Based on the scores assigned by the reviewers, we are happy to notify you that your paper was accepted for the workshop.

Please, address the comments of the reviewers and submit the camera-ready version by July 8. We ask the authors to record a 15min video for your talk. At the workshop, we will have the pre-recorded video as well as a live QA session. It is important to keep this time limit, otherwise, your talk will be automatically cut. The deadline for uploading the video is July 8. The detailed instructions for uploading will follow.

Feel free to contact us for any questions!

Best,

The ICML20@SaS organizers:
Mirco Ravanelli
Titouan Parcollet
Dmitriy Serdyuk
Devon Hjelm
Bhuvana Ramabhadran